# Association between Adverse Childhood Experiences and Multiple Sclerosis in Icelandic Women—A Population-Based Cohort Study

**DOI:** 10.3390/brainsci12111559

**Published:** 2022-11-16

**Authors:** Nicole M. Gatto, Edda Bjork Thordardottir, Gunnar Tomasson, Harpa Rúnarsdóttir, Huan Song, Jóhanna Jakobsdóttir, Thor Aspelund, Unnur Anna Valdimarsdóttir, Arna Hauksdóttir

**Affiliations:** 1Centre for Public Health Sciences, University of Iceland, 102 Reykjavik, Iceland; 2Department of Population and Public Health Sciences, Keck School of Medicine, University of Southern California, 1845 N Soto St., Los Angeles, CA 90032, USA; 3Mental Health Services, Landspitali, The National University Hospital of Iceland, 101 Reykjavik, Iceland; 4Landspitali, University Hospital, 101 Reykjavik, Iceland; 5West China Biomedical Big Data Center, West China Hospital, Sichuan University, Chengdu 610041, China

**Keywords:** trauma, multiple sclerosis, autoimmune, childhood, stressor, ACEs, adverse childhood experiences

## Abstract

Background: A growing literature, mostly based on selected populations, indicates that traumas may be associated with autoimmune diseases, yet few studies exist on adverse childhood experiences (ACEs) and multiple sclerosis (MS) in the general population. Objective: We assessed cross-sectional associations between self-reported ACEs and MS among Icelandic women in the population-based Stress-And-Gene-Analysis (SAGA) cohort. Methods: Participants (n = 27,870; mean age 44.9 years) answered a web-based survey that included the ACE-International Questionnaire and a question about MS diagnosis. Log-linear Poisson regression models estimated MS prevalence ratios and 95% confidence intervals for ACEs adjusted for covariates. Results: 214 women reported having been diagnosed with MS (crude prevalence = 7.7 per 1000). Compared to women without MS, women with MS reported more fatigue, body pain and bladder problems. The average cumulative number of ACEs was 2.1. After adjustment for age, education, childhood deprivation, smoking and depressive symptoms, MS prevalence did not increase with increasing ACEs exposure (PR = 1.00, 95% CI = 0.92, 1.09). Thirteen ACE categories, including abuse, neglect, household dysfunction and violence were not individually or independently associated with MS. Conclusion: Limited by self-reported data and cross-sectional design, results do not consistently support associations between ACEs in the development of MS among adult Icelandic women.

## 1. Introduction

The experience of traumas or other stressful life experiences is common among human populations globally. The 2001–2012 World Mental Health Surveys provide estimates that 70.4% of adults worldwide have experienced a trauma during their lifetime [1]. Traumas surveyed include war-related traumas, physical and sexual violence (including interpersonal and intimate partner violence), accidents (including natural disasters), unexpected death of a loved one, witness to or traumas among loved ones, and other forms of traumas. Data from the United States (US) for 2010 suggest 89.7% of US adults have ever experienced a traumatic event [2]. Compared with men, women are more likely to be exposed to interpersonal violence [2,3], and about equally as likely to experience an unexpected death of a loved one [3].

While traumatic and stressful experiences may be common, the effect of these experiences on mental and physical health varies. One well-studied consequence of trauma, posttraumatic stress disorder (PTSD), occurs to variable extents among affected populations, by sex and by event type [2,3]. Further studies have shown that traumatic and stressful experiences early in life are robustly associated with PTSD later in life [4]. Children with present or lifetime PTSD have cellular profiles indicating possible immune system dysregulation [5]. Early life adversities have been found to be associated with altered stress reactivity and regulation in infants, children and adolescents [6,7]. Animal studies have documented measurable changes on molecular, hormonal, behavioral, and social levels in response to stressors [8,9,10,11,12,13]. Taken together, this evidence suggests that epidemiologic studies should consider the effects of trauma and stress occurring at different ages on cognitive, emotional and behavioral disorders, as well as on physiological conditions. 

A growing literature exists among select populations showing that traumas and stressors may be associated with autoimmune (AI) diseases [14,15,16,17,18]. Active-duty US military personnel with PTSD enrolled in the Millennium Cohort Study had an elevated risk of multiple sclerosis (MS) and inflammatory bowel disease [14]. Veterans of the Iraq and Afghanistan wars under age 55 enrolled in the US Department of Veterans Affairs healthcare system with a PTSD diagnosis had a significantly higher risk of being diagnosed with any AI disorder, with risks being greater among women [19]. Male firefighters and emergency medical service workers involved with rescue/recovery efforts after the September 11 World Trade Center attack in New York had higher rates of systemic AI diseases, particularly systemic lupus erythematosus [20]. 

MS is an inflammatory AI disorder of the central nervous system (CNS) characterized by demyelination of neurons and white matter plaques in the brain. Clinical presentations of MS can include blurred vision with pain; impaired sensation, weakness and or loss of coordination; sensory disturbances; and other symptoms such as vertigo or hearing loss [21]. The etiology of MS is unknown though believed to result from interactions between genes and environmental factors [22], leading some to hypothesize that a sequential cascade of environmental insults occurring during childhood, adolescence and adulthood ultimately lead to its manifestation [23]. MS occurs twice as frequently in women than men, and predominantly affects persons of northern European ancestry [24]. 

Apart from research in selected traumatized populations, few epidemiologic studies have examined the effects of adverse childhood experiences (ACEs) on MS, and results have been mixed [25,26,27,28,29]. A 2011 study among women in the Nurses Health Cohort did not find an increased risk of MS with severe physical abuse during childhood or adolescence assessed retrospectively on questionnaire, but did observe non-statistically significantly elevated risks with being repeatedly forced into sexual activity in childhood and adolescence [27]. A 2012 clinic-based case–control study in Germany found that male and female patients with MS scored significantly higher on subscales for childhood emotional abuse and emotional neglect, sexual abuse, and severe abuse (either physical or sexual), but not on those for physical abuse or physical neglect on the Childhood Trauma Questionnaire compared with population controls [29]. In a cohort study of 2.9 million Danish men and women, exposure to any stressful life event before age 18 including parental divorce, parental death, or death of a sibling was associated with an 11% elevation in risk of MS [26]. Yet, in a nested case–control study among men prospectively recorded in the Swedish registry there was no difference in a derived stress resilience score for late adolescence between cases with inpatient and outpatient MS diagnoses and age-matched controls [25]. Taken together, the research on ACEs in MS is still underdeveloped and should continue to be explored.

In this cross-sectional cohort study among 27,870 Icelandic women, we examined whether the experience of ACEs is associated with the occurrence of MS during adulthood. We hypothesized that women who have greater exposure to ACEs would have a higher prevalence of MS compared with women who have less to no exposure to ACEs. We further explored whether associations with MS were more pronounced by categories of ACEs including forms of abuse, neglect, household dysfunction and violence.

## 2. Materials and Methods

### 2.1. Study Population

The Stress-And-Gene-Analysis (SAGA) study is a prospective cohort study conducted by the University of Iceland which aims to increase knowledge of the incidence of traumas and their effects on women’s health [30]. All Icelandic-speaking women 18–69 years of age and older residing in Iceland as of March 2018 (n = 104,197) were invited to participate. A total of 31,795 women were enrolled in the cohort and asked to complete an electronic questionnaire with sections on various traumatic experiences, and psychological and physical health. All questions were in Icelandic and response alternatives included “can’t/don’t want to answer”. The demographics of the participants were comparable to the Icelandic female population [31]. 

Of the 30,403 women who returned a questionnaire, the study was restricted to 27,870 SAGA participants who responded to an item enabling a categorization of MS and for whom no more than 50% of items on the Adverse Childhood Experience (ACE-IQ) were missing.

### 2.2. Measurements

The SAGA questionnaire included items on demographic factors (education, income, rural/urban residency, marital status), height and weight, current smoking status and alcohol use, medically diagnosed diseases, and current self-reported physical and mental health status. Childhood deprivation was assessed with the question, “Was your family’s economic situation ever so bad that you suffered deprivation? This refers to lack of nutritious food and/or warm clothes and appropriate footwear during the winter months”. Response options were never, rarely, sometimes and often. The Pittsburgh Sleep Quality Index (PSQI) [32] was used to assess sleep disturbances during the past month; the Patient Health Questionnaire-9 (PHQ-9) [33] was used to measure the severity of depressive symptoms during the past 2 weeks; and the Patient Health Questionnaire-15 (PHQ-15) [34] was used to assess somatic symptom severity during the past four weeks. Additional items queried current levels of fatigue and a physician diagnosis of neurocognitive disorder. Symptoms of posttraumatic stress disorder (PTSD) during the past month were assessed with the PTSD Checklist for DSM-5 (PCL-5) [35]. A validated cutoff score of 33 was used as the threshold for a probable diagnosis of PTSD [36].

### 2.3. Adverse Childhood Experiences (ACEs)

ACEs were measured with a modified version of the Adverse Childhood Experiences International Questionnaire (ACE-IQ) developed by the World Health Organization (WHO) [37]. The ACE-IQ consists of 39 items assessing how often individuals were exposed to 13 categories of ACEs during the first 18 years of their life: emotional or physical neglect by a parent or other household member, emotional or physical abuse by a parent or other household member, contact sexual abuse, domestic violence, living with a household member who abuses drugs and/or alcohol, living with a household member who is mentally ill or suicidal, incarceration of a household member, parental death or separation/divorce, being bullied, witnessing community violence, and exposure to war/collective violence (Appendix A). As collective violence such as wars are extremely rare in Iceland, the ACE-IQ was modified for use in SAGA by adding one screening question on collective violence. Response options varied between the 39 items and were either answered on a 5-point scale ranging from 0 (never) to 4 (always), a 4-point scale ranging from 0 (never) to 3 (many times), or dichotomously 0 (no) and 1 (yes). 

Scoring of the modified ACE-IQ followed the frequency approach according to the Guidance for Analyzing ACE-IQ [37]. In this approach, an affirmative response to each specific type of ACEs was documented if the required minimum frequency of events was met (Appendix A). For example, a woman was classified as having been physically abused in childhood if she answered “many times” on at least one of the two questions regarding physical abuse, while a woman was classified as having a history of sexual abuse if she “ever” experienced any of the 4 items enumerated for sexual abuse. We examined exposure to the 13 categories of ACEs individually (yes, no) as well as exposure to ACEs cumulatively. For the latter, the number of affirmative responses for each ACE was summed and then grouped into categories of 0, 1–2, ≥3 ACEs based on the distribution in the study population and following previous work [38]. 

### 2.4. Prevalent Multiple Sclerosis

Prevalent self-reported MS was determined by an affirmative response to the item “Have you ever been diagnosed with multiple sclerosis (MS)?”. To explore the validity of self-reported diagnosis of MS, we compared the prevalence of physical and cognitive health complaints or conditions which could reflect the occurrence of symptoms characteristic of MS, including pain, fatigue, and cognitive, sexual, bladder and bowel problems [21,39] between women in the SAGA cohort with and without MS. Using responses on the PHQ-15, we considered women to have experienced bodily pain including stomach pain, back pain or pain in the arms, legs or joints, if corresponding PHQ-15 items were reported as “bothered a lot” in the past month. Using the same reporting threshold and frequency for additional corresponding PHQ-15 items, we considered women to have sexual problems if they experienced pain or problems during sexual intercourse; and to have bowel conditions if they experienced constipation, loose bowels, or diarrhea. Based on responses on the PSQI, women who reported that getting up to use the bathroom 3 or more times per week in the past month was causing them to have trouble sleeping were considered to have bladder problems. Women who reported a physician-diagnosed neurocognitive disorder were categorized as such, and women who rated their fatigue during the last week as an 8 or higher out of 10 were considered to have prevalent fatigue. 

### 2.5. Covariates

Variables were categorized as follows: highest education level: primary, secondary education (high school or vocational education), college or equivalent (BSc or equivalent), and postgraduate (MSc or above); marital status: married/in a relationship and single/widowed; employment status: employed (including being a student and being on parental leave) and unemployed (including on disability benefits or on sick leave more than 2 months); monthly income [in thousands of US dollars (USD)]: ≤$2700, $2701–4500, ≥$4501 (conversion rates according to Central Bank of Iceland, October 17, 2018); body-mass index (BMI): <24.9, 25 to 29.9, and ≥30 kg/m^2^; smoking status: never smoked, former and current; number of alcoholic drinks per time on average: 0, 1–3, and ≥4 drinks.

### 2.6. Statistical Analysis

Descriptive statistics (means, frequencies) for demographic, lifestyle, anthropometric and self-reported health status characteristics were summarized and compared between women with and without MS using t-tests for continuous variables or chi-square tests for categorical variables. To ascertain the association between the cumulative number of ACEs and prevalent MS, log-linear Poisson regression models with robust error variance were used to estimate prevalence ratios (PRs) and 95% confidence intervals (CIs) [40]. Women who did not report a diagnosis of MS served as the reference group. We examined whether the association with MS was stronger among women with multiple ACEs. A base model was adjusted for age at the time of survey. A second model additionally adjusted for educational level and childhood deprivation given the known correlation of the latter with ACEs [41]. A third model included adjustment for a full set of covariates which included those selected for consideration because of their potential as risk factors for MS [42] or guided by previous research [25,26,27,28,29] including that on the SAGA cohort [43]. Covariables were retained in regression models if they modified associations by 10% or more or if there was a rationale for their inclusion. In sensitivity analyses, we excluded 36 women who reported a neurocognitive disorder because of the potential effect on recall of questionnaire items. 

We also explored the association between MS and the 13 individual categories of ACEs including types of abuse, neglect, household dysfunction and violence. Models were adjusted for age at the time of the survey, educational level, childhood deprivation level, smoking status and current depression symptoms. Because of the interrelatedness of ACEs [44], in a second set of models, we mutually adjusted for all other categories of ACEs to examine the independent association between each category of ACE and MS. 

Owing to the inverse association between age and trauma in the SAGA cohort (younger women report more trauma) and since MS is typically diagnosed between the ages of 20–30 years with some diagnostic delay in Iceland [45] (we expect some cases of MS have not yet been diagnosed in young women in the cohort), we restricted analyses of the cumulative number of ACEs to women older than 30 years. Since ACEs are correlated with psychiatric disorders [46], we examined whether the association between the cumulative number of ACEs and MS was modified by PTSD by including a multiplicative interaction term between ACEs and PTSD in a fully adjusted model. 

Statistical tests were two-sided with a significance level set 5%. All analyses were conducted in R Version 4.1.2 using RStudio (2 September 2021). 

The SAGA study was approved by the National Bioethics Committee of Iceland (no. VSNb2017110046/03.01) and the Icelandic Data Protection Authority. All participants gave informed consent before participation.

## 3. Results

A total of 214 women of 27,870 in the SAGA cohort reported having ever been diagnosed with MS, for a crude prevalence of 7.7 per 1000. Women with MS were comparable to women in the cohort without MS with respect to age, relationship status, highest level of education, place of birth (Iceland or another country), residence in an urban or rural area, alcohol consumption, smoking status, BMI and childhood deprivation levels. Compared to women without MS, women with MS were somewhat more likely to report experiencing depressive symptoms of greater intensity during the past two weeks and PTSD symptoms during the past month, but the differences were not statistically significant. Women with MS had statistically significantly lower monthly incomes and were less likely to be actively working, relative to women without MS (Table 1).

The average cumulative number of ACEs reported by women overall was 2.1 (SD: 2.2); the number was similar between women with MS (mean: 2.2, SD: 2.3) and those without (mean 2.1, SD 2.2). Nearly one-third of women without MS reported experiencing three or more ACEs, a proportion that was slightly higher among women with MS (35.2%) (Table 2). Regression models did not support an association between ACEs overall or a greater number of ACEs and MS prevalence. After adjustment for age, education, childhood deprivation, smoking and depressive symptoms, the PR was 1.00 (95% CI = 0.92, 1.09) (Table 2). Women who had experienced three or more ACEs were also no more likely to have MS (PR = 0.99; 95% CI = 0.63, 1.54). In analyses restricted to women older than 30 years, there was no association between cumulative number of ACEs and MS (PR = 0.99, 95% CI = 0.90, 1.07). There was also no evidence of effect modification by PTSD with the *p*-value for the interaction term = 0.71.

Several categories of ACEs were reported frequently by women in the SAGA cohort. Over one-quarter experienced some form of household dysfunction in childhood, with parental separation or divorce being the most common (Table 3). Nearly a third of women reported that a parent or guardian rarely or never understood their problems or worries or knew what they were doing with their free time. The experience of community violence, collective violence, living with an incarcerated household member, and physical or sexual abuse were the least prevalent. Family violence and bullying were reported more frequently by women with MS than by women without MS (Table 3). In multivariable regression models, no specific category of ACE was statistically significantly individually associated with MS. Prevalence ratios were greater than the null for six of the 13 categories of ACEs including emotional and sexual abuse, physical neglect, family violence, parental separation or divorce and bullying. Prevalence ratios were less than the null for five categories including emotional neglect, substance abuse, incarcerated household member, mental illness and community violence, and were null for physical abuse and collective violence (Table 3). 

When mutually adjusted for other categories, no ACE was statistically significantly independently associated with MS (Table 3). Among the 13 categories of ACEs, the independent associations with MS were the strongest between physical neglect (PR = 1.32, 95% CI = 0.71, 2.32), parental separation or divorce (PR = 1.21, 95% CI = 0.86, 1.68) and bullying (PR = 1.27, 95% CI = 0.83, 1.91). Because of findings from a previous study [27], we estimated prevalence ratios for specific forms of sexual abuse. While the confidence intervals were wide and included the null, the associations with MS were stronger with more severe forms of sexual abuse: forced into touching someone else in a sexual way (PR = 0.90, 95% CI = 0.35, 1.89); unwanted touching or fondling by someone in a sexual way (PR = 1.25, 95% CI = 0.69, 2.11); unwanted attempt of oral, anal or vaginal intercourse (PR = 1.34, 95% CI = 0.52, 2.82); forced into oral, anal or vaginal intercourse (i.e., rape) (PR = 1.40, 95% CI = 0.49, 3.12).

In exploring the validity of a self-reported MS diagnosis, we found that women with MS were more likely to report experiencing fatigue during the previous week compared to women without MS (38.3% versus 24.3%, *p* < 0.0001), and to have body pain (52.8% versus 46.3%, *p* = 0.05) and bladder problems (36.0% versus 26.7%, *p* = 0.007) during the past month, while the occurrence of sexual problems and bowel conditions were not different between women with and without MS. Neurocognitive disorders were extremely rare among SAGA women (Table 4) and only one woman with MS reported having received such a diagnosis from a physician. Unsurprisingly, therefore, the estimated associations did not change when women who reported a neurocognitive disorder were excluded from analyses.

## 4. Discussion

In this study of 27,870 adult Icelandic women in the SAGA cohort, we found that exposure to a greater number of adverse experiences during childhood was not associated with MS reported during adulthood. Generally, the categories of ACEs did not follow any consistent pattern of being individually or independently correlated with MS during adulthood. Prevalence ratios for physical neglect, parental separation or divorce, and bullying were elevated, yet with confidence intervals that included the null. 

Of the previous research, the two prospective registry-based studies [25,26] incorporated objective, albeit limited, measures of stressors during childhood and adolescence. The nationwide study in Denmark obtained registrations of divorce between parents and death of a parent or sibling to identify stressful life events prior to age 18 [26]. The Swedish study among male military conscripts used a measure of stress coping ability determined from a psychological examination conducted at ages 18 and 19 years [25]. While our measurement of childhood traumas and other stressors was through self-report, it provided much greater detail than either of these studies. We relied on women to report if they had been diagnosed with MS, while the Danish study identified MS cases from the national MS registry and the Swedish study from the national patient register, both using ICD codes. The former study found that exposure to one stressful life event was associated with an 10% increased risk for developing MS among women, and for men and women together, this was primarily due to an experience of divorce rather than death. The latter study found no association between stress resilience and MS. Two other previous epidemiologic studies with sex-specific results retrospectively assessed childhood stressors of abuse and neglect via questionnaire [27,28]. Among women in the Nurses’ Health Study, physical or sexual abuse during childhood or adolescence, regardless of the level of severity, were not associated with self-reported incident MS diagnoses verified by neurologists as definite or probable. Neither was physical abuse or physical neglect associated with MS in a German case–control study which enrolled cases of definite MS from a university outpatient clinic and community-based controls. However, female MS patients were twice as likely as controls to report emotional neglect or sexual abuse and over 3 times as likely to report emotional abuse. 

Studying risk factors for complex chronic diseases like MS is challenging. This is due, in part, to the complicated nature of addressing timing of and interactions between multiple contributing factors, if one follows the sequential cascade of insults hypothesis [23]. Applying a life course approach to our results, it could be that women in SAGA accumulated other advantages such as higher levels of education or income, which acted to offset or buffer potential later life health effects from negative childhood experiences [47]. 

A limitation of this study is the use of self-reported MS diagnosis which could introduce measurement error as it may be less valid than physician-confirmed diagnosis (i.e., from medical records or hospital linkage) and could reflect over-reporting by women. Reviews of records from MS patient registries in the US and the United Kingdom, respectively, have found that a high proportion (98%) of self-reported MS diagnoses can be verified [48] and that data from a registry population are comparable to those identified from clinical specialty centers [49]. However, the crude MS prevalence calculated among SAGA women who provided a response to the questionnaire item is higher than that estimated [50] for the female Icelandic population in 2007 which only included index cases with diagnostic verification from medical record review who met defined diagnostic criteria. Eliasdottir et al. [50] reported age-specific prevalence rates among Icelandic women highest in the 40–44-, 45–49- and 50–54-year age groups at 583, 446 and 526 per 100,000, respectively, relative to all but the 60–64-year age group. Given the SAGA cohort was restricted to women 18–69 years of age and the mean age of women in our analysis was 45 years, we expect MS prevalence in our sample to be proportional to the age distribution of enrolled women. Yet, concerns about lack of diagnostic validation persist and future studies in the SAGA cohort will be strengthened by a linkage to hospital records. In the current study, we find the higher rates of fatigue, pain, bladder problems, and depression among women with self-reported MS compelling, as these are among the most common symptoms of the disease [21,39]. Taken together, these symptoms help to substantiate the self-reports of MS diagnoses and reduce some concerns of measurement error. Additionally, the absence of diagnostic validation is unlikely to have led to biases in the prevalence ratios, specifically to missing an association between MS and ACEs. 

Furthermore, while questionnaire data on exposure (ACEs) and outcome (MS) were obtained cross-sectionally, we are less concerned about temporal ambiguity because the focus on adverse experiences during childhood would precede MS onset which typically occurs between the ages of 20 and 40 years [23]. Reports of ACEs were collected retrospectively, yet retrospective responses to childhood abuse and related household dysfunction have been shown to be generally reliable over time [51]. Nevertheless, among women with prevalent MS, it is not possible to rule out recall bias. The SAGA study did not include direct assessment of cognitive function, and thus we cannot exclude the possibility that reporting of ACEs was impacted by cognitive impairment, which could have been more frequent or of greater severity among women with MS [22]. Yet, completion of the SAGA questionnaire would have required a level of cognitive ability to be able to do so, and physician-diagnosed neurocognitive disorders were extremely rare in study participants. The SAGA questionnaire did not collect information about age of MS diagnosis, so we also cannot discount instances (although expected to be infrequent [22]) of MS with a pediatric onset. This could have implications in our analysis for the experiences of trauma among women occurring prior to or coinciding with MS onset.

We explored thirteen different categories of ACEs, and a larger sample size may be needed to clarify possible associations with physical neglect, parental separation or divorce and bullying. Because of previous reports [27] and our observation of stronger associations for more severe forms of sexual abuse, a closer examination of childhood sexual abuse in MS by future studies that include validated cases of definite MS is merited. Additionally, research into environmental factors related to potentially higher MS prevalence in Nordic countries [50] may consider other factors that may have regional variation such as exposure to traumatic events [1]. 

A strength of this project is the large population cohort providing good power to examine associations. This basis of the study in a general population cohort of women is additionally important because the types of stressful and traumatic events are more likely reflective of those experienced by the Icelandic female population. Therefore, our results have greater generalizability than previous work among active military, veterans, first responders or emergency medical personnel [14,15,16,17,18]. However, the results of this study are not generalizable to Icelandic men. Another strength of the study is the psychological instruments which are embedded within the SAGA study questionnaire (PHQ-9, PHQ-15, PSQI, ACE-IQ) are all validated measures. The ACE-IQ is a comprehensive assessment of childhood stressors including traumas, providing more detailed data than that of previous studies [25,26,27,28]. 

In summary, using a large population-based female cohort, we generally did not observe cross-sectional associations between adverse childhood experiences and MS. This result is to some extent supported by previous studies using prospective designs. The implications of this research are important for at least two reasons. For one, given the widespread experience of traumas or other stressful life experiences among human populations and the prevalence of ACEs among women in our study, our findings will be reassuring in that they suggest that greater exposure to traumas and stressors during childhood or specific categories of ACEs do not contribute to the development of MS. Second, our results are meaningful to future research, particularly with the widespread interest in ACEs and their role in human health. While ACEs have been firmly documented as risk factors for numerous other diseases, a null finding may guide the scientific community to consider possible mechanisms for ACE-disease associations when results bear out selectively for some diseases and not others.

## Figures and Tables

**Table 1 brainsci-12-01559-t001:** Characteristics of women with and without prevalent multiple sclerosis (MS) in the SAGA Cohort, number (%) or mean ± standard deviation.

Characteristic	No MS (N = 27,656)	MS (N = 214)
Age (years)		
Mean ± SD	44.9 ± 14.2	45.6 ± 11.1
18–30	5029 (18.2%)	15 (7.0%)
30–39	5365 (19.4%)	60 (28.0%)
40–49	6025 (21.8%)	58 (27.1%)
50–59	6263 (22.6%)	58 (27.1%)
60–69	4974 (18.0%)	23 (10.7%)
Relationship status		
In a relationship	20,768 (75.1%)	169 (79.0%)
Single and/or widowed	6742 (24.4%)	43 (20.1%)
Unknown	146 (0.5%)	2 (0.9%)
Highest Level of Education		
Primary/Elementary school	4094 (14.8%)	33 (15.4%)
Secondary ^a^	8571 (31.0%)	64 (29.9%)
University degree	14,875 (53.8%)	116 (54.2%)
Unknown	116 (0.4%)	1 (0.5%)
Monthly Income (USD) *		
≤$2700	8274 (29.9%)	85 (39.7%)
$2701–4500	8378 (30.3%)	72 (33.6%)
≥$4501	9918 (35.9%)	49 (22.9%)
Unknown	1086 (3.9%)	8 (3.7%)
Job Status †		
Actively working	22,774 (82.3%)	118 (55.1%)
Not actively working	4692 (17.0%)	94 (43.9%)
Unknown	190 (0.7%)	2 (0.9%)
Born in Iceland		
No	1171 (4.2%)	10 (4.7%)
Yes	26,432 (95.6%)	204 (95.3%)
Unknown	53 (0.2%)	0 (0%)
Residence in urban or rural area		
Urban	25,386 (91.8%)	198 (92.5%)
Rural	1823 (6.6%)	14 (6.5%)
Unknown	447 (1.6%)	2 (0.9%)
Number of alcoholic drinks per time, on average		
none	2686 (9.7%)	25 (11.7%)
1	5152 (18.6%)	42 (19.6%)
2–3	11,765 (42.5%)	96 (44.9%)
≥4	6691 (24.2%)	39 (18.2%)
Unknown	1362 (4.9%)	12 (5.6%)
Smoking status		
Never	12,983 (46.9%)	101 (47.2%)
Former	10,136 (36.7%)	84 (39.3%)
Current	4194 (15.2%)	27 (12.6%)
Unknown	343 (1.2%)	2 (0.9%)
BMI		
≤24.9	9681 (35.0%)	73 (34.1%)
25–29.9	9031 (32.7%)	64 (29.9%)
≥30	8604 (31.1%)	75 (35.0%)
Unknown	340 (1.2%)	2 (0.9%)
Depressive symptoms ^b^		
minimal	10,871 (39.3%)	68 (31.8%)
mild	8589 (31.1%)	68 (31.8%)
moderate	4375 (15.8%)	44 (20.6%)
moderately severe	2472 (8.9%)	20 (9.3%)
severe	1349 (4.9%)	14 (6.5%)
Childhood Deprivation level		
never	20,843 (75.4%)	159 (74.3%)
seldom	3070 (11.1%)	26 (12.1%)
sometimes	2451 (8.9%)	15 (7.0%)
often	1202 (4.3%)	14 (6.5%)
Unknown	90 (0.3%)	0 (0%)
PTSD		
No	25,558 (92.4%)	193 (90.2%)
Probable	2098 (7.6%)	21 (9.8%)

BMI, body mass index; PTSD, posttraumatic stress disorder. * *p* < 0.001; † *p* < 0.0001. ^a^ Includes high school, vocational, trade school. ^b^ During past 2 weeks.

**Table 2 brainsci-12-01559-t002:** Associations between number of and cumulative exposure to adverse childhood experiences (ACEs) and multiple sclerosis (MS) among women in the SAGA Cohort.

	No MS Mean ± SD or Number (%)	MS Mean ± SD or Number (%)	Model 1 PR (95% CI) ^a^	Model 2 PR (95% CI) ^b^	Model 3 PR (95% CI) ^c^
Total number of ACEs, per ACE	2.1 ± 2.2	2.2 ± 2.3	1.03 (0.96, 1.10)	1.01 (0.93, 1.09)	1.00 (0.92, 1.09)
By number of ACEs					
0	5972 (26.8)	44 (26.7)	Ref.	Ref.	Ref.
1–2	9033 (40.6)	63 (38.2)	0.95 (0.65, 1.40)	0.94 (0.64, 1.39)	0.94 (0.64, 1.41)
≥3	7259 (32.6)	58 (35.2)	1.09 (0.74, 1.62)	1.02 (0.67, 1.56)	0.99 (0.63, 1.54)

PR, prevalence ratio; CI, confidence interval. ^a^ Adjusted for age at the time of the survey. ^b^ Additionally adjusted for educational level, childhood deprivation level. ^c^ Additionally adjusted for smoking status, current depression level.

**Table 3 brainsci-12-01559-t003:** Prevalence of and associations between individual categories of adverse childhood experiences (ACEs) ^a^ and multiple sclerosis (MS), SAGA Cohort.

ACE	No MS, Number (%)	MS, Number (%)	PR (95% CI) ^b^	PR (95% CI) ^c^
Abuse				
Physical	1484 (5.4%)	13 (6.1%)	1.02 (0.54, 1.76)	0.51 (0.17, 1.23)
Emotional	4300 (15.5%)	37 (17.3%)	1.10 (0.74, 1.60)	1.13 (0.65, 1.91)
Sexual	1278 (4.6%)	13 (6.1%)	1.24 (0.66, 2.12)	1.04 (0.46, 2.04)
Neglect				
Physical	2139 (7.7%)	20 (9.3%)	1.13 (0.67, 1.81)	1.32 (0.71, 2.32)
Emotional	8843 (32.0%)	68 (31.8%)	0.89 (0.64, 1.22)	0.81 (0.54, 1.20)
Household dysfunction				
Family violence	6858 (24.8%)	62 (29.0%)	1.13 (0.81, 1.55)	1.15 (0.73, 1.77)
Parental separation or divorce	10,630 (38.4%)	89 (41.6%)	1.11 (0.83, 1.49)	1.21 (0.86, 1.68)
Substance abuse	9527 (34.4%)	67 (31.3%)	0.82 (0.6, 1.11)	0.89 (0.60, 1.29)
Incarcerated household member	1541 (5.6%)	11 (5.1%)	0.89 (0.45, 1.58)	0.92 (0.40, 1.83)
Mental illness	8802 (31.8%)	66 (30.8%)	0.85 (0.62, 1.16)	0.95 (0.65, 1.38)
Violence				
Community violence	1052 (3.8%)	7 (3.3%)	0.81 (0.34, 1.62)	0.62 (0.19, 1.54)
Bullying	4633 (16.8%)	48 (22.4%)	1.17 (0.88, 1.57)	1.27 (0.83, 1.91)
Collective violence	1128 (4.1%)	10 (4.7%)	1.02 (0.48, 1.89)	1.13 (0.47, 2.26)

N, number; PR, prevalence ratio; CI, confidence interval; PRs compare women with exposure to the individual category of ACE to those without (reference). ^a^ Prevalence of Adverse Childhood Experiences defined using the frequency version of the ACE-IQ. ^b^ Adjusted for age at the time of the survey, educational level, childhood deprivation level, smoking status and current depression level. ^c^ Additionally adjusted for other categories of ACEs.

**Table 4 brainsci-12-01559-t004:** Prevalence [number (%)] of self-reported physical and cognitive health conditions among women with and without MS in the SAGA Cohort.

	No MS	MS	
	N = 27,656	N = 214	*p*-Value ^a^
Body pain ^b,c^	46.3%	52.8%	0.05
Fatigue	24.3%	38.3%	<0.0001
Neurocognitive disorder ^d^	0.1%	0.5%	0.22
Bladder problems ^b^	26.7%	36.0%	0.007
Sexual problems ^b,e^	5.0%	3.3%	0.29
Bowel conditions ^b,f^	21.1%	23.8%	0.33

^a^ adjusted for age at the time of the survey, educational level, childhood deprivation level. ^b^ during last month. ^c^ includes stomach pain, back pain or pain in the arms, legs or joints, last 4 weeks. ^d^ physician-diagnosed. ^e^ pain or problems during sexual intercourse, last 4 weeks. ^f^ constipation, loose bowels, or diarrhea, last 4 weeks.

## Data Availability

The data used in this study are compiled in the Stress-And-Gene-Analysis (SAGA) cohort. We cannot make the data publicly available because of Icelandic laws regarding data protection and the approval for the current study granted by the National Bioethics Committee (NBC) of Iceland. The SAGA cohort contains sensitive data and all use of data is restricted to scientific purposes only subjected to approval of the NBC (email: vsn@vsn.is). Interested researchers can obtain access to deidentified data by submitting a proposal to the SAGA cohort data management board (email: afallasaga@hi.is) which assists with submitting an amendment to the NBC. The corresponding author of the present study submitted a research proposal to the SAGA cohort data management board; the NBC and was granted access only to deidentified data, which cannot be shared.

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
