# Peer review of "Association between Adverse Childhood Experiences and Multiple Sclerosis in Icelandic Women—A Population-Based Cohort Study"

_brainsci, 2022, doi:10.3390/brainsci12111559_

Round 1
Reviewer 1 Report
1. The manuscript presents the relationship between adverse childhood experiences with the occurrence of MS during adulthood. It is well written, although in my opinion has a limited clinical value. Below are some comments regarding the text:
-The title should include data on the name of the geographical region in which the study was conducted.
- As the study applies to female, conclusions in abstract should strictly apply to the investigated group.
- The study is based on a self-reported MS diagnosis which is associated with few limitations. For example, some patients may have early onset/paediatric MS and therefore the disease may have originated before experiencing traumatic event.
- The prevalence of MS in the studied cohort differs significantly from the available data (see DOI: 10.1159/000489472) - please comment.
- The incidence of traumatic events can be very geographic-region dependent, as well as the incidence of MS - so do the authors see any potential correlation in this issues?
Author Response
Reviewer #1
- The title should include data on the name of the geographical region in which the study was conducted.
Response: Thank you for this suggestion. We have modified the title to indicate the study was conducted among Icelandic women.
- As the study applies to female, conclusions in abstract should strictly apply to the investigated group.
Response: We appreciate the reviewer’s request to make this distinction. We have added to the abstract to specify that the conclusion applies to adult Icelandic women.
- The study is based on a self-reported MS diagnosis which is associated with few limitations. For example, some patients may have early onset/paediatric MS and therefore the disease may have originated before experiencing traumatic event.
Response: The reviewer raises an important point about pediatric onset MS which could lead to reverse causation in our study. We have added a discussion if this in lines 388-392.
- The prevalence of MS in the studied cohort differs significantly from the available data (see DOI: 10.1159/000489472) - please comment.
Response: Thank you for calling out the Eliasdottir et al 2018 study. We comment on how our calculated prevalence compares with those from that research in the discussion section of our revised manuscript. We have also revised our text to emphasize that the MS prevalence is our study is crude.
- The incidence of traumatic events can be very geographic-region dependent, as well as the incidence of MS - so do the authors see any potential correlation in this issues?
Response: The reviewer raises an interesting point. We comment on this as an area of potential further investigation in lines 397-400 of this revised manuscript.

Reviewer 2 Report
The authors wrote a retrospective study aimed to investigate the association between multiple sclerosis and adverse childhood experiences.
The cohort is large, but I have many major concerns regarding methodology:
- the authors found a MS prevalence higher respect to other studies carry out in Iceland: Elíasdóttir Ó et al found a 167/100000 prevalence, while the authors found a 770/100000 prevalence. This finding should be deeply analyzed and commented, because it rises the doubt of a biased estimate.
- MS diagnosis was self reported, so there is a high likelihood of detection bias. In fact the prevalence of MS is higher when compared to a recent study
- The study design is cross sectional, the survey is aimed to investigate the history of childhood, the age distribution is quite different between MS and non-MS (women under 30 yers and over 60 years are more than double in non-MS) and memory function could be impaired in MS women. Those findings rose the doubt of recall bias about adverse childhood experiences.
- neuro-cognitive impairment afflicts 40-70% oh MS woman, but in this study the prevalence in 0.5%, and there is no significant (p 0.22) difference compared to non-MS women: these finding could reflect a selection bias (people with cognitive impairment could not be able to answer to the survey) and an ascertainment bias (how was cognitive impairment ascertained?)
For those reason I have robust doubt regarding biased estimates in this study.
Author Response
Reviewer #2
- The authors found a MS prevalence higher respect to other studies carry out in Iceland: Elíasdóttir Ó et al found a 167/100000 prevalence, while the authors found a 770/100000 prevalence. This finding should be deeply analyzed and commented, because it rises the doubt of a biased estimate.
Response: We thank the reviewer for their concern which was shared by Reviewer #1. As noted in our response above, we address the Eliasdottir et al study in our revised manuscript (lines 360-367). We note that the 167 per 100,000 prevalence which the reviewer cites from Eliasdottir et al is for all ages and both sexes.
- MS diagnosis was self reported, so there is a high likelihood of detection bias. In fact the prevalence of MS is higher when compared to a recent study.
Response: Thank you. As noted above, we address that there could be over-reporting by women in our study. We have strengthened our acknowledgement of the absence of diagnostic validation in our study and point to future plans for hospital linkage (lines 368-370).
- The study design is cross sectional, the survey is aimed to investigate the history of childhood, the age distribution is quite different between MS and non-MS (women under 30 yers and over 60 years are more than double in non-MS) and memory function could be impaired in MS women. Those findings rose the doubt of recall bias about adverse childhood experiences.
Response: We appreciate the reviewer’s concern. Women in SAGA were a maximum of 69 years to be eligible for enrollment and, women without MS who were 60-69 years old were proportionally greater than women with MS. We have revised Table 1 to show upper and lower bounds of the age groups to reflect the cohort inclusion criteria. We have addressed recall bias in the discussion, lines 382-383. As we note below, our study did not directly assess cognitive function and was thus not able to classify women as cognitively impaired or not.
- neuro-cognitive impairment afflicts 40-70% oh MS woman, but in this study the prevalence in 0.5%, and there is no significant (p 0.22) difference compared to non-MS women: these finding could reflect a selection bias (people with cognitive impairment could not be able to answer to the survey) and an ascertainment bias (how was cognitive impairment ascertained?)
Response: We thank the reviewer for raising this important point. Our study did not directly assess cognitive function and was thus not able to classify women as cognitively impaired or not. We direct the reviewer to our methods section, line 177 where we describe that we queried for a physician-diagnosed neurocognitive disorder. It is true that women participating in SAGA would have required a level of cognitive ability to complete the questionnaire. We have added this to the discussion, line 386-387.

Reviewer 3 Report
Your paper shows women in general are more at risk to experience past (in this work ACEs) emotional, psychological and physical violence traumas, while being more prone as well to be affected by past stressors of diverse nature. These findings however have not been consistent in the literature as you appropriately report. While your study did not show cross-sectional associations between adverse childhood experiences and MS, (even though their possible pathophysiological mechanisms in the MS process remains to be established), your work provides important information and should incite interest into future studies. One different approach to consider would be exploring in the cohort of women with already established definite MS, the risk of emotional and psychological traumas exerted by abuse from spouse, significant other or care giver. Difficult topic to investigate.
Author Response
Your paper shows women in general are more at risk to experience past (in this work ACEs) emotional, psychological and physical violence traumas, while being more prone as well to be affected by past stressors of diverse nature. These findings however have not been consistent in the literature as you appropriately report. While your study did not show cross-sectional associations between adverse childhood experiences and MS, (even though their possible pathophysiological mechanisms in the MS process remains to be established), your work provides important information and should incite interest into future studies. One different approach to consider would be exploring in the cohort of women with already established definite MS, the risk of emotional and psychological traumas exerted by abuse from spouse, significant other or care giver. Difficult topic to investigate.
Response: We thank the reviewer for their comments. We added to our recommendation that future studies (lines 396-397) include cases of validated definite MS.

Round 2
Reviewer 1 Report
All suggested changes are included in the text of the manuscript. I have no additional comments.
Reviewer 2 Report
the authors addressed my concerns.